# Incidence and Risk Factors of COVID-19-Associated Pulmonary Aspergillosis in Intensive Care Unit—A Monocentric Retrospective Observational Study

**DOI:** 10.3390/pathogens10111370

**Published:** 2021-10-22

**Authors:** Emilien Gregoire, Benoit François Pirotte, Filip Moerman, Antoine Altdorfer, Laura Gaspard, Eric Firre, Martial Moonen, Vincent Fraipont, Marie Ernst, Gilles Darcis

**Affiliations:** 1Department of Internal Medicine and Infectious Diseases, Centre Hospitalier Régional (CHR) de Liège, 4000 Liège, Belgium; benoit.pirotte@chrcitadelle.be (B.F.P.); filip.moerman@chrcitadelle.be (F.M.); antoine.altdorfer@chrcitadelle.be (A.A.); laura.gaspard@chrcitadelle.be (L.G.); eric.firre@chrcitadelle.be (E.F.); martial.moonen@chrcitadelle.be (M.M.); 2Intensive Care Unit, Centre Hospitalier Régional (CHR) de Liège, 4000 Liège, Belgium; vincent.fraipont@chrcitadelle.be; 3Biostatistics and Medico-Economic Information Department, Centre Hospitalier Universitaire (CHU) de Liège, 4000 Liege, Belgium; m.ernst@chuliege.be; 4Department of Internal Medicine and Infectious Diseases, Centre Hospitalier Universitaire (CHU) de Liège, 4000 Liège, Belgium; gdarcis@chuliege.be

**Keywords:** coronavirus disease 2019 (COVID-19), invasive pulmonary aspergillosis (IPA), COVID-19-associated pulmonary aspergillosis (CAPA), intensive care unit (ICU), bronchoalveolar lavage (BAL), galactomannan (GM)

## Abstract

Coronavirus disease 2019 (COVID-19)-associated pulmonary aspergillosis (CAPA) is an increasingly recognized complication of COVID-19 and is associated with significant over-mortality. We performed a retrospective monocentric study in patients admitted to the intensive care unit (ICU) for respiratory insufficiency due to COVID-19 from March to December 2020, in order to evaluate the incidence of CAPA and the associated risk factors. We also analysed the diagnostic approach used in our medical centre for CAPA diagnosis. We defined CAPA using recently proposed consensus definitions based on clinical, radiological and microbiological criteria. Probable cases of CAPA occurred in 9 out of 141 patients included in the analysis (6.4%). All cases were diagnosed during the second wave of the pandemic. We observed a significantly higher realization rate of bronchoalveolar lavage (BAL) (51.1% vs. 28.6%, *p* = 0.01) and *Aspergillus* testing (through galactomannan, culture, PCR) on BAL samples during the second wave (*p* < 0.0001). The testing for *Aspergillus* in patients meeting the clinical and radiological criteria of CAPA increased between the two waves (*p* < 0.0001). In conclusion, we reported a low but likely underestimated incidence of CAPA in our population. A greater awareness and more systematic testing for *Aspergillus* are necessary to assess the real incidence and characteristics of CAPA.

## 1. Introduction

Critically ill patients with acute respiratory distress syndrome (ARDS) caused by viral pneumonia are at higher risk of bacterial and fungal co-infections, including invasive pulmonary aspergillosis (IPA) [1]. Severe influenza is known as a risk factor of IPA in intensive care unit (ICU) patients [2,3] with the incidence of influenza associated with pulmonary aspergillosis (IAPA) reported as between 7% and 28% worldwide [2,4,5,6,7,8,9]. Notably, there are continental variations in the reported incidence rates, with rates ranging from 16% to 23% in Europe [2,7,8,9], 17% to 28% in Asia [5,6] and as low as 7% in Alberta, Canada [4].

Coronavirus disease 2019 (COVID-19), caused by the severe acute respiratory syndrome coronavirus 2 (SARS-CoV-2) infection, emerged in December 2019 and has since become a pandemic. It may cause severe pneumonia with ARDS, requiring intensive care and mechanical ventilation (MV), which has a high mortality rate [10,11]. Due to its similarity with influenza and ARDS, severe COVID-19 is suspected to be a potential risk factor for IPA [12,13]. COVID-19 patients are also considered as susceptible to opportunistic fungal superinfections due to lymphopenia and the defective function of lymphocytes, as well as the systemic cytokine hyperinflammatory reaction, which are associated with COVID-19 [13]. Finally, since the generalisation of dexamethasone (DXM) use for the treatment of COVID-19 patients receiving respiratory support from August 2020 [14], immunodepression due to corticosteroids [15] became an additional risk factor of superinfections in these patients [16], as seen in influenza [17].

Cases of COVID-19-associated aspergillosis (CAPA) were reported early and in large numbers [18,19,20,21,22]. The reported incidence rate of CAPA among ICU-hospitalized COVID-19 patients varies widely, ranging from 2.4% to 35% [7,23]. Two main multicentre prospective cohort studies on CAPA, by Bartoletti et al. (108 patients) [24] and White et al. (135 patients) [25], reported a CAPA incidence rate of 27.7% and 14.1%, respectively. These variations in reported incidence may be due to several factors: host factors (ethnicity and genomic factors), environmental factors (the facilities’ ventilation systems, construction material, and nearby constructions) and variations in the practitioner’s and institution’s diagnostic definitions and approaches (including galactomannan (GM) testing and bronchoscopy realization), influenced by infection control policies [7,23].

CAPA is a concern due to its associated over-mortality [7]. The prospective studies of Bartoletti et al. and White et al. both identified a significant excess mortality rate of 25% (44% vs. 19%) [24] and 27% (58% vs. 31%) [25] in the CAPA group compared to the control group without IPA. The patients with CAPA often cumulate several prognostic factors of mortality, among those identified in 2019 by Koehler et al. in patients with invasive aspergillosis (IA) [26]. In their recent review of the literature on CAPA, Pasquier et al. identified high age, chronic pulmonary disease and elevated serum GM as independent risk factors of mortality in CAPA patients in a multivariate analysis [7]. High age and pulmonary disease are also risk factors of mortality in the general COVID-19 population, but elevated serum GM might reflect a real influence of CAPA on mortality [27]. Similarly, in their recent meta-analysis, Singh et al. reported an overall mortality rate in CAPA patients of 52.2% [28]. However, unlike Pasquier et al., their meta-regression analysis did not reveal any association between mortality in CAPA and any risk factor, COVID-19-specific therapy or anti-fungal therapy [28].

Identifying the risk factors of developing CAPA is necessary to help identify and prevent it. The most important risk factors identified are the use of corticosteroids, presence of comorbidities such as structural lung defects, severe lung damage during the course of COVID-19, and the use of broad-spectrum antibiotics [13,17,18,19,20,22,29]. White et al. identified chronic respiratory disease and use of corticosteroids as independent risk factors of developing CAPA. [25]. Bartoletti et al. identified chronic corticosteroids treatment as a risk factor of CAPA [24]. These risk factors are similar to those of influenza-associated pulmonary aspergillosis (IAPA) [2]. Further studies are needed to determine which corticosteroids and which posology represent a risk of developing CAPA.

The diagnosis of CAPA, and, more broadly, of IPA in critically ill patients, is challenging and debated [30]. The IPA case definition and classification of the European Organization for Research and Treatment of Cancer and the Mycosis Study Group Education and Research Consortium (EORTC/MSG) [31], used for neutropenic and immune-compromised patients, is not suitable for critically ill patients in the ICU and non-immune-compromised patients. It is too restrictive, and thus lacks sensitivity, as these patients often lack the host criteria and have a non-specific imaging pattern associated with ARDS and their underlying respiratory infections [1,18,19,24]. In addition, qualitative mycological evidence from the respiratory samples is needed to ensure the distinction between the colonization and the true IPA and to avoid over-diagnosing CAPA [30]. However, obtaining such samples is challenging in these patients, due to clinical instability, hypoxia and the risk of aerosol contamination of healthcare workers [32]. To address this matter and appropriately define and classify CAPA, several case definitions have been proposed for IPA in critically patients (AspICU study) [1], for IAPA (modified AspICU) [2], and now for CAPA [3,24,30]. The European Confederation for Medical Mycology and the International Society for Human and Animal Mycology (ECMM/ISHAM) recently proposed a consensus definition for CAPA, based on a review of the available literature [33]. Here, we used the definition proposed by Verweij et al. and Koehler et al. [3,33], which was the most recent definition and seemed the most adapted and consensual at the time that this study was conducted.

We reported the incidence of CAPA in the two intensive care units of our medical centre, from the 1 March 2020 through 31 December 2020. We studied the risk factors of developing CAPA. Finally, we evaluated the diagnostic approach used in our centre for CAPA diagnosis since the beginning of the pandemic, including the frequency of the realization of bronchoalveolar lavage (BAL) and microbiological tests on blood and BAL samples, and its influence on CAPA epidemiology.

## 2. Materials and Methods

### 2.1. Design and Participants

We conducted a retrospective observational study. We included only adult (≥18 years) patients admitted between 1 March 2020 and 31 December 2020 to the two ICUs of our regional hospital for respiratory insufficiency due to COVID-19 and diagnosed with a laboratory-confirmed SARS-CoV-2 infection (by real-time polymerase chain reaction (RT-PCR) via nasopharyngeal swab or endotracheal aspirate (ETA) or BAL) anytime during 2 weeks between hospital admission and ICU admission, or positive RT-PCR within 72–96 h after ICU admission. Patients who were transferred to another facility during their ICU stay and whose evolution and outcome could not be known were excluded.

The patient data were anonymously collected and included the following points: demographical characteristics; medical history and co-morbidities; hospital and ICU length of stay; survival; COVID-19 microbiological tests (PCR or enzyme-linked immunosorbent assay serology); COVID-19 treatment (DXM, tocilizumab, siltuximab, anakinra, convalescent plasma, remdesivir, azithromycin and/or hydroxychloroquine); respiratory sampling (sputum, ETA and BAL); processing for *Aspergillus* culture, GM and *Aspergillus* PCR on those respiratory samples (sputum, ETA or BAL) and GM and *Aspergillus* PCR on blood samples; chest X-ray (CXR) or computed tomography (CT) scan; antibiotic treatment; and antifungal treatment (voriconazole, isavuconazole, echinocandin, amphotericin B).

CAPA cases were classified as proven, probable or possible CAPA depending on their completion of clinical, radiological and microbiological criteria, according to the definitions recently proposed by Koehler et al. [33].

The region amplified to detect *Aspergillus by PCR* in both BAL and blood was the 87-base pair ITS2 region of the 18S rRNA gene.

The study was conducted according to the guidelines of the Declaration of Helsinki and approved on the 16 August 2021 by the Ethics Committee of the CHR Citadelle of Liège (412), under protocol code JL/bl/TFE2021/09-E.GREGOIRE - B4122021000029. 

### 2.2. Statistics

Categorical variables were described using frequency tables, while continuous quantitative variables were described using statistical summaries (mean, standard deviation, minimum and maximum, median and interquartile range). 

Simple logistic regression models were used to identify risk factors. For each model, the Odd Ratio (OR), 95% confidence intervals (CI) and p-values were reported. If the ORs from the simple logistic regressions were not directly calculable, a Haldane correction was performed, and the p-value of the Fisher exact test was provided. In a second step, a multivariate logistic model was used to identify risk factors for CAPA diagnosis. Variables with an individual p-value below the threshold of 0.10 were added to the model.

Survival was modelled using a Kaplan–Meier curve and was compared between the two groups using the log-rank test. The chi-square test (or Fisher’s exact test in case of small numbers) was used to compare the proportions between two groups. The results were considered significant at the 5% uncertainty level (*p* < 0.05).

Calculations were made using SAS (SAS Institute, Cary, NC, USA) version 9.4 and graphs using R (R Foundation for Statistical Computing, Vienna, Austria) version 3.6.1. 

## 3. Results

Over the period, from 1 March 2020 through 31 December 2020, 141 patients were admitted to one of the two ICUs of our hospital with a confirmed positive SARS-CoV-2 infection and respiratory insufficiency due to COVID-19. The demographic characteristics and comorbidities of the study population are described in the Appendix A, along with respiratory support and COVID-19 treatment administered (Appendix A).

Using the ECMM/ISHAM definition, the incidence of CAPA in our population was 6.4%, with 9/141 patients meeting the criteria for probable CAPA. Using the modified AspICU definition, the same nine patients (6.4%) met the definition of putative IPA. Another patient (0.7%) was classified as *Aspergillus* colonisation, as he did not meet any clinical criteria. The median time from ICU admission to CAPA diagnosis was 15 days (min = 0 days; Q1 = 10 days; Q3 = 15 days; max = 29 days). Seven of the nine (7/9) CAPA cases were treated with voriconazole, and one with isavuconazole. One case was not treated. All nine probable CAPA cases occurred during the second wave of the COVID-19 epidemic in our region (after 1 August 2020). The remaining 132 patients had no criteria for CAPA according to these two definitions. The diagnostic criteria, treatment received and outcome of the nine probable CAPA cases are described in Table 1.

The risk factors for CAPA in the multivariate analysis are described in Table 2. The univariate analysis is available in the Appendix A. Being diagnosed in second wave was the only risk factor associated with CAPA in the multivariate analysis (OR > 999, *p* = 0.011). No demographical characteristic was significantly associated with CAPA. A medical history of cerebrovascular disease (OR = 6.83, *p* = 0.078) and arterial hypertension (OR = 7.53, *p* = 0.052), as well as respiratory support by MV (OR = 13, *p* = 0.070), were associated with a trend towards an increased risk of CAPA. Immunosuppressive treatment, treatment by azithromycin (AZT) and/or hydroxychloroquine (HCQ) and by dexamethasone (DXM) were all associated with a trend towards an increased risk of CAPA in the univariate analysis, but no statistically relevant association was identified in the multivariate analysis.

Regarding the diagnoses, we observed that the rate of BAL performed as well as GM. Furthermore, *Aspergillus* PCR and *Aspergillus* cultured performed on BAL were significantly higher during the second wave than the first (BAL realization: 51.1% vs. 28.6%, *p* = 0.01). This was even more significant in the subgroup of patients combining clinical and radiological criteria (86% vs. 36.7%, *p* < 0.0001) (Table 3). Furthermore, the rate of BAL performed was higher in the group of patients with both clinical and radiological criteria of CAPA compared to patients without these two criteria, but this was significant only in the second wave (86% vs. 9.5%, *p* < 0.001) and not in the first wave (36.7% vs. 15.8%, *p* = 0.11) (Table 4).

The mortality in the group of probable CAPA was higher than in non-CAPA patients, but it did not reach a statistical significance (Figure 1, panel A, B). When analysing patients included in the second wave only, the statistical difference was stronger, but still not statistically significant (Figure 2, panel A, B). Comparing the two waves, we observed that mortality was higher during the first wave than during the second wave. This difference was not statistically significant, although a trend was observed (*p* < 0.10) (Appendix A in the Appendix A). In addition, mortality occurred later in the second wave (P25 = 28.5 days) than in the first wave (P25 = 10 days) (Appendix A in the Appendix A).

## 4. Discussion

The incidence of probable CAPA among ICU hospitalized COVID-19 patients in our study (6.4%) was low compared to the literature. The two main multicentre prospective cohort studies on CAPA reported CAPA incidence rates of 27.7% and 14.1% [24,25]. One possible reason for this low incidence is that we have a monocentric cohort, so our sample population may be less representative of the general population of COVID-19 patients in intensive care. The incidence of CAPA can also vary considerably depending on specific local demographic or environmental characteristics. [7,23]. The burden of *Aspergillus* in the environment of our hospital’s ICU could be low, leading to a lower rate of *Aspergillus* superinfection in our patients. The definition used for CAPA diagnosis could also influence the reported incidence [7,24,25]. We used the consensus definition proposed by the ECMM/ISHAM expert panel [33]. It allowed the diagnosis of IPA to be made on the basis of non-specific radiological criteria and microbiological criteria that were more sensitive than culture alone. Moreover, incidence in our study was the same when using this definition or the former definition of modified AspICU [2], suggesting the adequateness of the definition’s criteria. The definition used was also the same than the one used in the prospective studies of White et al. and Bartoletti et al. [24,25]. For these reasons, we do not believe that the definition that we used explains the low incidence of CAPA in our study. Interestingly, we observed a trend of the higher and earlier mortality of COVID-19 patients in the ICU during the first wave than during the second wave. This better outcome during the second wave could be explained, at least in part, by the use of DXM as a standard of care in COVID-19 treatment since the publication of the RECOVERY study [14]. Therefore, it could be possible that, during the first wave, some patients did not live long enough to have had time for CAPA to develop and/or to be diagnosed. Finally, the incidence of CAPA was obviously influenced by local protocols regarding sampling and testing procedures and strategies [7,23]. We indeed observed a significantly higher incidence of CAPA in the second wave compared to the first wave, with the nine probable CAPA cases diagnosed during the second wave. We also showed a significantly higher realization rate of BAL and testing (culture, GM, PCR) on BAL samples in the second wave. This difference was even more significant in patients who combined clinical and radiological CAPA criteria. This suggests that the awareness of CAPA in our institution increased during the pandemic and that the diagnostic strategy evolved towards more frequent respiratory diagnostic investigations during the second wave. Nonetheless, in spite of the increased awareness, 14% of patients of the second wave with both clinical and radiological criteria still did not have BAL performed, and *Aspergillus* cultures, GM, and *Aspergillus* PCR in BAL were not processed in 18%, 24% and 28% of these patients, respectively. Therefore, we believe that CAPA was likely under-researched, especially in the first wave, and that the true incidence of CAPA may have been underestimated in our population.

We identified a trend towards an association between MV and CAPA. Interestingly, Bartoletti et al. studied the incidence of CAPA exclusively in intubated patients [24]. As critical COVID-19 patients inherently require mechanical ventilation for respiratory support [34], the severity of COVID-19 could be an important confounder. Indeed, these patients often have ARDS, which is characterised by pulmonary epithelial damage and temporary immune deregulation, and which could favour *Aspergillus* invasion alone [13,34]. However, the severity of the disease was not documented in our population, and thus was not analysed in our study; this should be analysed in future prospective studies.

We did not observe a statistically significant association between CAPA and any comorbidity or medical history. Only the history of cerebrovascular disease and hypertension showed a trend associated with CAPA. Bartoletti et al. [24] observed in their prospective study a similar trend for cerebrovascular disease but not for hypertension. We did not observe any association between chronic obstructive pulmonary disease or other chronic lung diseases and CAPA, in contrast to what White et al. reported [25]. Immunosuppressive treatment, as well as treatment by AZT and/or HCQ and by DXM, showed a trend towards an association with CAPA in univariate analysis, but no association in the multivariate analysis.

Interestingly, AZT was previously identified as a risk factor for CAPA in another retrospective study [34]. AZT is known to have an immunomodulatory effect, by inhibiting neutrophils and innate immune responses, and thus could reduce immune defence against *Aspergillus* [34]. Additionally, its broad-spectrum antibiotic effect could alter the microbiota of patients, thereby promoting *Aspergillus* colonisation [34]. There may therefore be a link between AZT treatment and CAPA, but this needs to be further investigated.

Corticosteroids were suspected to be early risk factors for CAPA, based on their pharmacological effects and on previous experience in other severe pulmonary viral infections [15,17]. In an early retrospective study, Dellière et al. observed a trend towards an association between CAPA and a high cumulative dose (>100 mg) of DXM-equivalent [34]. Later, both Bartoletti et al. [24] and White et al. [25] identified corticosteroid therapy (as chronic therapy and as a COVID-19 treatment, respectively) to be an independent risk factor for CAPA. In our study, we observed a trend towards a higher probability of CAPA in DXM-treated patients in the univariate analysis, but no association was observed between DXM and CAPA in the multivariate analysis. It is important to note that the dose of DXM used in our patients (maximum 60 mg cumulated) was lower than the >100 mg of DXM equivalent described by Dellière et al. 

The ECMM published in August 2021 a broad, multicentric, multinational observational study on 592 COVID-19 patients from twenty centers in nine countries [35]. In their multivariate analysis, invasive ventilation, older age and treatment with tocilizumab were significantly associated with the increased probability of CAPA development, but there was no significant association with systemic corticosteroid therapy [35]. The differences between this study and our study could be explained in part by the larger size of their population sample and the multicentric design of their study. Particularly regarding tocilizumab, only two patients were treated with this molecule in our population, which prevented us from making any significant observations about this treatment.

Recent systematic reviews and a meta-analysis of CAPA reported an overall mortality rate of 51.2% [28], 52.2% [7] and 54.9% [36]. The prospective studies of Bartoletti et al. and White et al. reported the mortality rates of 44% vs. 19% [24] and 58% vs. 31% [25] 30 days after ICU admission, in the CAPA group compared to the non-CAPA control group, respectively. This represents a significant excess mortality in the CAPA group of 25% and 27%, respectively [24,25]. The mortality rate in the CAPA group in our study was 55.6% (5/9), which is close to the data in these reviews. Compared to the 43.9% (58/132) mortality rate in the non-CAPA group, the mortality rate in the CAPA group was higher, although this was not statistically significant.

Our study has several limitations. First, it has a retrospective design. Moreover, due to the relatively small sample size of our population and the low incidence of CAPA, it is likely underpowered for the use in creating an extensive multivariate model on risk factors. This is displayed by the fact that no predictor was significant in the multivariate model. In addition, we studied our population as a whole and the heterogeneity of the population was not assessed. Yet, many parameters may have changed between the two waves and throughout the pandemic. For example, the standard of care in treating COVID-19 evolved from AZT and HCQ in the first wave, to DXM in the second wave, after the Randomised Evaluation of COVID-19 Therapy (RECOVERY) trial results were published [14]. Other factors, some of which may not yet have been identified, may have changed between the two waves and thus induced a bias for certain factors concerning the risk of CAPA. 

In conclusion, we report a low but likely underestimated incidence of CAPA in our population. We show that the awareness and more systematic search for *Aspergillus* are necessary to assess the real incidence of CAPA. Prospective studies with a systematic screening for IPA are needed to better define the incidence and risk factors of CAPA, ideally comparing patients with COVID-19 to those with other viral pneumonias, at the same time and in the same intensive care units.

## Figures and Tables

**Figure 1 pathogens-10-01370-f001:**
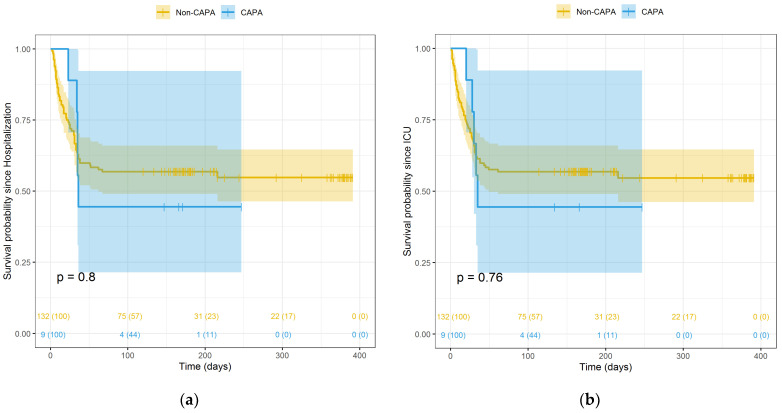
Kaplan–Meier survival curves of CAPA vs. non-CAPA groups throughout study, since hospitalization admission (**a**) and since ICU admission (**b**).

**Figure 2 pathogens-10-01370-f002:**
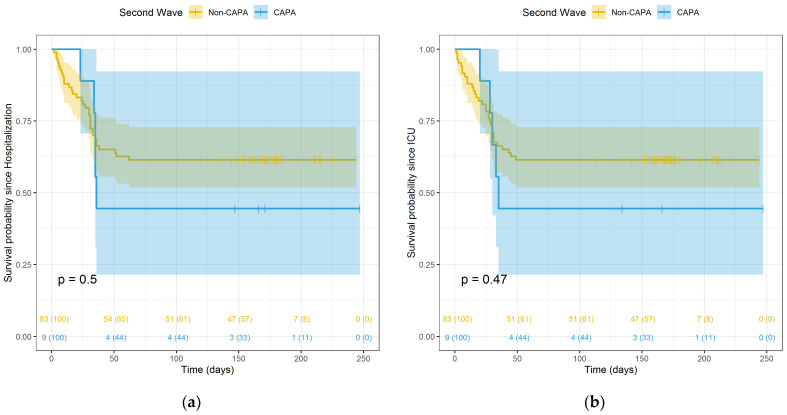
Kaplan–Meier survival curves of CAPA vs. non-CAPA groups in the second wave, starting from hospitalization admission (**a**) and from ICU admission (**b**).

**Table 1 pathogens-10-01370-t001:** Characteristics of CAPA patients.

Case Number	Wave	Clinical Criterion	Compatible CXR Sign	Compatible CT Sign	GM on BAL	GM on ETA	GM on Blood	*Aspergillus* PCR on BAL	*Aspergillus* PCR on Blood	*Aspergillus* Culture on BAL	*Aspergillus* Culture on ETA	*Aspergillus* Culture on Sputum	CAPA Diagnosis (ECMM/ ISHAM)	IPA Diagnosis (Modified AspICU)	CAPA Treatment	Outcome
1	2nd	Fever ^1^ Respiratory degradation	Condensating infiltrate	NA	1.2	NA	NA	Negative	NA	Negative	NA	NA	Probable CAPA	Putative IPA	Voriconazole	Alive
2	2nd	Fever ^1^ Respiratory degradation	Condensating infiltrate	NA	1.5	NA	<0.5	Positive ^2^ (Qualitative result)	NA	Negative	Positive ^2^	NA	Probable CAPA	Putative IPA	Voriconazole	Dead
3	2nd	Respiratory degradation	Condensating infiltrate	NA	4.5	NA	<0.5	Negative	NA	Positive ^2^	Positive ^2^	NA	Probable CAPA	Putative IPA	Isavuconazole	Dead
4	2nd	Respiratory degradation	Condensating infiltrate	NA	3.3	NA	NA	Negative	NA	Negative	NA	NA	Probable CAPA	Putative IPA	Voriconazole	Alive
5	2nd	Respiratory degradation	Condensating infiltrate	Condensating infiltrate	2.7	NA	0	Negative	NA	Negative	NA	NA	Probable CAPA	Putative IPA	Voriconazole	Dead
6	2nd	Fever ^1^ Respiratory degradation	Condensating infiltrate	NA	4.5	NA	NA	Negative	Negative	Negative	NA	NA	Probable CAPA	Putative IPA	Voriconazole	Alive
7	2nd	Fever ^1^ Respiratory degradation	Condensating infiltrate	NA	3.4	NA	NA	Negative	NA	Negative	NA	NA	Probable CAPA	Putative IPA	Not treated	Dead
8	2nd	Fever ^1^ Respiratory degradation	Condensating infiltrate	Condensating infiltrate	2.7	NA	<0.5	Negative	NA	Negative	NA	Positive ^2^	Probable CAPA	Putative IPA	Voriconazole	Alive
9	2nd	Respiratory degradation	Condensating infiltrate	NA	1.5	NA	NA	Positive ^2^ (Qualitative result)	NA	Negative	NA	NA	Probable CAPA	Putative IPA	Voriconazole	Dead

^1^ Refractory fever despite at least 3 days antibiotics or recrudescent fever of at least 48 hours despite appropriate antibiotherapy. ^2^ Positive for *Aspergillus fumigatus*. CXR = Chest X-ray; CT = computed tomography; GM = Galactomannan; PCR = Polymerase chain reaction; BAL = bronchoalveolar aspiration; ETA = endotracheal aspiration; CAPA = COVID-19-associated pulmonary aspergillosis; IPA = invasive pulmonary aspergillosis ; ECMM/ISHAM = European Confederation for Medical Mycology and the International Society for Human and Animal Mycology; NA = not assessed.

**Table 2 pathogens-10-01370-t002:** Risk factors of CAPA—multivariate model.

			Univariate		Multivariate		
Variable	Categories	OR	95% CI	*p*-Value	OR	95% CI	*p*-Value
Demographical variables							
Age (years)		1.05	0.98–1.12	0.14			
BMI (kg/m²)		0.97	0.86–1.10	0.66			
Sex	Women	1.33	0.32–5.62	0.69			
Wave	2nd wave	11.3	0.62–203	0.10	>999	6.35->999	0.121
Comorbidities							
Hypertension	Yes	7.30	0.89–60.1	0.064	7.53	0.99–57.4	0.052
Cerebrovascular disease	Yes	6.00	1.02–35.3	0.048	6.83	0.80–58.0	0.078
Diabetes	Yes	1.50	0.38–5.84	0.56			
Thrombo-embolic disease	Yes	1.02	0.04–24.6	0.99			
COPD	Yes	2.41	0.46–12.7	0.30			
Former TB	Yes	2.75	0.06–120	0.60			
Former Aspergillosis	Yes	4.68	0.05–447	0.51			
HIV	Yes	2.75	0.06–120	0.60			
Obesity	Yes	0.3	0.06–1.57	0.16			
Cardiopathy	Yes	0.73	0.15–3.69	0.71			
Smoking				0.92			
	Former vs. Non	0.53	0.02–11.8				
	Active vs. Non	0.92	0.23–3.62				
Alcoholism				0.54			
	Former vs. Non	1.24	0.05–32.3				
	Active vs. Non	3.15	0.41–24.2				
Other pneumopathy	Yes	0.97	0.19–4.93	0.97			
CKD	Yes	1.53	0.17–13.4	0.70			
Hepatopathy	Yes	0.51	0.03–10.4	0.66			
Neoplasia	Yes	0.56	0.03–11.5	0.70			
Malignant hemopathy	Yes	4.00	0.40–40.1	0.24			
Benign hemopathy	Yes	8.13	0.66–99.4	0.10			
Auto-immune disease	Yes	1.38	0.16–12.0	0.77			
Immunodeficiency	Yes	2.75	0.06–120	0.60			
Immunosuppressive treatment	Yes	5.10	0.89–29.2	0.067	5.43	0.67–44.0	0.11
Treatment							
Respiratory support	MV vs. Other	12.8	0.71–230	0.084	13.0	0.81–206.3	0.070
Antibiotic at admission	Yes	4.12	0.22–77.8	0.35			
AZT and HCQ	AZT and/or HCQ	0.088	0.01–1.58	0.099	41.1	0.27->999	0.15
Remdesivir	Yes	1.22	0.05–31.1	0.90			
Dexamethasone	Yes	11.6	0.64–210	0.097	0.29	0.003–29.3	0.60
Tocilizumab	Yes	2.75	0.06–120	0.60			
Siltuximab	Yes	4.68	0.05–447	0.51			
Anakinra	Yes	1.50	0.05–42.1	0.81			
Convalescent plasma	Yes	3.18	0.33–30.5	0.32			
Other corticosteroids	Yes	1.95	0.06–63.5	0.71			
Antibiotic during hospitalization	At least one	0.82	0.03–20.9	0.90			

AZT = azithromycin; BMI = body mass index; CI = confidence interval; CKD = chronic kidney disease; COPD = chronic obstructive pulmonary disease; HCQ = Hydroxychloroquine; HIV = human immunodeficiency virus; MV = mechanical ventilation; OR = odds ratio; TB = tuberculosis.

**Table 3 pathogens-10-01370-t003:** Comparison of realization rate of BAL and of GM, *Aspergillus* PCR and *Aspergillus* cultures on BAL between the two waves in the whole population sample (**A**) and in the subgroup of patients with both clinical and radiological criteria (**B**).

**A. Population Sample (N = 141)** **Test Realised**	**1st Wave (N = 49)** **N (%)**	**2nd Wave (N = 92)** **N (%)**	** *p* ** **-Value**
BAL	14 (28.6)	47 (51.1)	0.010
BAL GM	7 (14.3)	42 (46.6)	0.0002
BAL *Aspergillus* PCR	7 (14.3)	40 (43.5)	0.0005
BAL *Aspergillus* culture	13 (26.5)	45 (48.9)	0.010
**B. Patients with both Clinical and** **Radiological Criteria (N = 80)** **Test Realised**	**1st Wave (N = 30)** **N (%)**	**2nd Wave (N = 50)** **N (%)**	** *p* ** **-Value**
BAL	11 (36.7)	43 (86.0)	<0.0001
BAL GM	6 (20.0)	38 (76.0)	<0.0001
BAL *Aspergillus* PCR	6 (20.0)	36 (72.0)	<0.0001
BAL *Aspergillus* culture	10 (33.3)	41 (82.0)	<0.0001

BAL = bronchoalveolar aspiration; GM = Galactomannan; PCR = Polymerase chain reaction.

**Table 4 pathogens-10-01370-t004:** Comparison of realization rate of BAL and of GM, *Aspergillus* PCR and *Aspergillus* cultures on BAL between patients with both clinical and radiological criteria of CAPA and patients without both criteria, in the first wave (**A**) and the second wave (**B**).

**A. First Wave (N = 49)** **Clinical and Radiological Criteria** **Test Realised**	**No (N = 19)** **N (%)**	**Yes (N = 30)** **N (%)**	** *p* ** **-Value**
BAL	3 (15.8)	11 (36.7)	0.11
BAL GM	1 (5.3)	6 (20.0)	0.15
BAL *Aspergillus* PCR	1 (5.3)	6 (20.0)	0.15
BAL *Aspergillus* Culture	3 (15.8)	10 (33.3)	0.18
**B. Second Wave (N = 92)** **Clinical and Radiological Criteria** **Test Realised**	**No (N = 42)** **N (%)**	**Yes (N = 50)** **N (%)**	** *p* ** **-Value**
BAL	4 (9.5)	43 (86.0)	<0.0001
BAL GM	4 (9.5)	38 (76.0)	<0.0001
BAL *Aspergillus* PCR	4 (9.5)	36 (72.0)	<0.0001
BAL *Aspergillus* Culture	4 (11.5)	41 (82.0)	<0.0001

BAL = bronchoalveolar aspiration; GM = Galactomannan; PCR = Polymerase chain reaction.

## Data Availability

The data presented in this study are available on request from the corresponding author.

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
