# Peer review of "Incidence and Risk Factors of COVID-19-Associated Pulmonary Aspergillosis in Intensive Care Unit—A Monocentric Retrospective Observational Study"

_pathogens, 2021, doi:10.3390/pathogens10111370_

Round 1
Reviewer 1 Report
This is an interesting study on incidence and risk factors of COVID-19-associated pulmonary aspergillosis in intensive care unit form a single center involving 141 patients of which 9 were diagnosed with CAPA. The main limitations are the low sample size, and results need to be put into context taking into account recently published much larger studies. That said, the manuscript is well written and of interest.
Specific comments:
- Please discuss how risk factors and outcomes differ from the much larger multicentric ECMM CAPA study (recently first papers published in CMI (PMID: 34454093) and Intens Care Med (PMID: 34269853). This is particularly relevant as this other study included results from 4 other centers from Belgium. Please discuss.
- The sample size of 9 cases is likely underpowered for doing an extensive multivariate model on risk factors. This is also displayed by the fact that no predictor was significant in the multivariate model. Please comment in limitations.
Author Response
Point 1: Please discuss how risk factors and outcomes differ from the much larger multicentric ECMM CAPA study (recently first papers published in CMI (PMID: 34454093) and Intens Care Med (PMID: 34269853). This is particularly relevant as this other study included results from 4 other centers from Belgium. Please discuss.
Response 1: These very recent publications are indeed particularly relevant, and it is very interesting to discuss our results in the light of this study.
The ECMM published in August 2021 a wide multicentric multinational observational study on 592 COVID-19 patients from 20 centers in 9 countries. In their multivariate analysis, invasive ventilation, older age and treatment with tocilizumab were significantly associated with increased probability of CAPA development, but there was no significant association with systemic corticosteroid therapy. The differences between this study and ours could be explained in part by the bigger size of their population sample and the multicentric design of their study. Regarding tocilizumab in particular, only two patients were treated with this molecule in our population, which prevents us from making any significant observation about this treatment.
We have adapted the discussion in the revised manuscript as requested.
Point 2: The sample size of 9 cases is likely underpowered for doing an extensive multivariate model on risk factors. This is also displayed by the fact that no predictor was significant in the multivariate model. Please comment in limitations
Response 2: We do agree with this remark from reviewer 1 and believe that the low statistical power of our study, due to our small sample size and the relatively low incidence of CAPA in our population, is one of the limitations of our study. The limitation paragraph of the discussion has been adapted as follows to express this limitation more clearly and precisely.
“Our study has several limitations. First, it has a retrospective design. Moreover, due to the relatively small sample size of our population and the low incidence of CAPA, our study is likely underpowered for doing an extensive multivariate model on risk factors. This is displayed by the fact that no predictor was significant in the multivariate model.”

Reviewer 2 Report
The manuscript clearly and adequately addresses a topic of great interest to the epidemiological situation currently being experienced worldwide. However, I have some observations.
Material and methods
It is convenient to mention which region was amplified to detect Aspergillus in BAL and blood.
Results
Table 1. In the column "Aspergillus PCR on BAL", what does “unknown viral load” refer to?
In patients with PCR or positive culture for Aspergillus, what were the species found?
Author Response
Point 1: Material and methods: It is convenient to mention which region was amplified to detect Aspergillus in BAL and blood.
Response 1: The region amplified to detect Aspergillus in both BAL and blood was the 87-base pair ITS2 region of the 18S rRNA gene. This information regarding the method used for Aspergillus PCR is indeed interesting and useful. We have added it in the part "2. Materials and Methods - Design and participants" of the revised manuscript.
Point 2: Results: Table 1. In the column "Aspergillus PCR on BAL", what does “unknown viral load” refer to?
Response 2: This is a sensible question and we recognize that the term used may have been confusing. "Unknown viral load" meant that the PCR was answered as a qualitative result, without providing a viral load nor a cycle threshold. To be more accurate, we have adapted the "Table 1. Characteristics of CAPA patients" in the part "3. Results" of the revised manuscript and replaced "Unknown viral load" by "qualitative result".
Point 3: Results: In patients with PCR or positive culture for Aspergillus, what were the species found?
Response 3: The species found was Aspergillus fumigatus in each case.
- Case 2: PCR on BAL positive for Aspergillus fumigatus. ETA culture positive for Aspergillus fumigatus.
- Case 3: BAL and ETA culture positive for Aspergillus fumigatus.
- Case 8: Sputum culture positive for Aspergillus fumigatus.
- Case 9: PCR on BAL positive for Aspergillus fumigatus.
We have adapted the "Table 1. Characteristics of CAPA patients" in the part "3. Results" of the revised manuscript. We have included a referral ("2") in this table and a legend mentioning "2 Positive for Aspergillus fumigatus" in the legend of the table.
